# KPT330 improves Cas9 precision genome- and base-editing by selectively regulating mRNA nuclear export

Yan-ru Cui[1,8], Shao-jie Wang[1,8], Tiancheng Ma[2], Peihong Yu[1,3], Jun Chen[4], Taijie Guo[2], Genyi Meng[2], Biao Jiang[1], Jiajia Dong [2✉] & Jia Liu [1,5,6,7✉]

CRISPR-based genome engineering tools are associated with off-target effects that constitutively active Cas9 protein may instigate. Previous studies have revealed the feasibility of modulating Cas9-based genome- and base-editing tools using protein or small-molecule CRISPR inhibitors. Here we screened a set of small molecule compounds with irreversible warhead, aiming to identifying small-molecule modulators of CRISPR-Cas9. It was found that selective inhibitors of nuclear export (SINEs) could efficiently inhibit the cellular activity of Cas9 in the form of genome-, base- and prime-editing tools. Interestingly, SINEs did not function as direct inhibitors to Cas9, but modulated Cas9 activities by interfering with the nuclear export process of Cas9 mRNA. Thus, to the best of our knowledge, SINEs represent the first reported indirect, irreversible inhibitors of CRISPR-Cas9. Most importantly, an FDA-approved anticancer drug KPT330, along with other examined SINEs, could improve the specificities of CRISPR-Cas9-based genome- and base editing tools in human cells. Our study expands the toolbox of CRISPR modulating elements and provides a feasible approach to improving the specificity of CRISPR-Cas9-based genome engineering tools.

[1] Shanghai Institute for Advanced Immunochemical Studies and School of Life Science and Technology, ShanghaiTech University, 201210 Shanghai, China. [2] Key Laboratory of Organofluorine Chemistry, Center for Excellence in Molecular Synthesis, Shanghai Institute of Organic Chemistry, Chinese Academy of Sciences, 345 Ling-Ling Road, 200032 Shanghai, China. [3] University of Chinese Academy of Sciences, 100049 Beijing, China. [4] College of Life Sciences, Zhejiang University, 310058 Hangzhou, Zhejiang, China. [5] Shanghai Clinical Research and Trial Center, 201210 Shanghai, China. [6] Gene Editing Center, School of Life Science and Technology, ShanghaiTech University, 201210 Shanghai, China. [7] Guangzhou Laboratory, Guangzhou International Bio Island, No. 9 XingDaoHuanBei Road, 510005 Guangzhou, Guangdong, China. [8] These authors contributed equally: Yan-ru Cui, Shao-jie Wang. ✉email: jiajia@sioc.ac.cn; liujia@shanghaitech.edu.cn

Clustered Regularly-Interspaced Short Palindromic Repeats (CRISPR) and CRISPR-associated (Cas) proteins (CRISPR-Cas) are the bacterial immune system for defending bacteriophage infection[1]. Type II CRISPR contains a single streamlined nuclease that can be reprogrammed for various genome engineering applications[2–4]. In human cells, CRISPR-Cas9 can induce DNA double-strand breaks that are repaired by two competing mechanisms: non-homologous end joining and, in the presence of DNA templates, homology-directed repair[5]. Therapeutic genome editing often involves constitutively expressed Cas9 protein[6], which may introduce excessive double-strand breaks and error-prone non-homologous end joining, leading to off-target mutations[7–9], chromosomal rearrangement[10], or genotoxicity[11].

Recently, CRISPR-derived base editors[12] (BEs) have been developed to overcome the adverse effects associated with CRISPR-based genome editing tools[13]. BEs are fusion proteins comprising a catalytically inactive Cas nuclease, a nucleobase deaminase and, in some cases, DNA glycosylase inhibitors[12,14]. BEs can generate nucleotide substitutions without introducing double-strand breaks or DNA template[12] and are thus optimal choice for precision gene therapy[15,16]. Similar to genome editing tools, uncontrolled BEs are also found to be associated with high frequency of off-target events[17–19]. This problem is particularly recognized in cytosine base editors (CBEs) in comparison with relatively high-fidelity adenine base editors (ABEs)[20,21].

To mitigate these side effects, temporal control-enabling CRISPR-Cas inhibitors have been investigated. Thus far, naturally occurring, phage-derived anti-CRISPR proteins (Acrs) are the most characterized CRISPR inhibitors. A number of Acrs have been identified for type I[22–26], type II[27–29], and type V[26,30] CRISPR-Cas systems. These inhibitors exert their functions by disrupting distinct steps of CRISPR-Cas actions such as single-guide RNA (sgRNA) binding[31], DNA binding[32] or DNA cleavage[32]. In nature, Acrs are used by bacteriophages to escape the CRISPR-Cas immunity in bacteria[22,33]. In genome engineering applications, Acrs are adapted to modulate CRISPR-Cas functions in a variety of host cells including bacteria[34], yeast[35] and mammalian cells[27,32,34,36,37]. In addition, Acrs can be coupled with CRISPR-Cas system to construct biosensors[38] or synthetic circuits in eukaryotes[39]. The ability of Acrs to achieve temporal and spatial[40] or optogenetic[41] control of CRISPR-Cas9 has enabled their applications for improving the targeting specificity[37], cell type-dependent activity[42] and cytotoxicity[11,43] of CRISPR-based genome editing tools.

In addition to protein-based inhibitors, oligonucleotides[44] and phage-derived peptides[31] have also been developed as CRISPR off-switches. Importantly, small-molecule inhibitors of CRISPR-Cas9 have been identified using an in vitro high-throughput screening assay[45]. These small molecules can reversibly inhibit the cellular activity of CRISPR-Cas9 by disrupting Cas9-DNA interactions[45]. Despite of the myriad types and mechanisms, these CRISPR-Cas inhibitors remain poorly understood for their therapeutic potential, particularly the safety in human.

In the present study, we used an EGFP reporter-based live cell assay to screen for irreversible small-molecule inhibitors of CRISPR-Cas9. We discovered that selective inhibitors of nuclear export (SINEs), including a marketed anti-cancer drug KPT330 (selinexor), could regulate the cellular activity of CRISPR-Cas9 by interfering with the nuclear export of Cas9 mRNA. We subsequently found that these SINEs could be used to modulate the activity and specificity of CRISPR-Cas9-based genome editing, base editing and prime editing tools.

## Results

### Screening for irreversible small-molecule inhibitors of CRISPR-Cas9. 
While most existing CRISPR-Cas inhibitors exert their functions via reversible, non-covalent interactions, we speculated that irreversible inhibitors could be also suitable options for CRISPR-Cas9 that is exogenous to human cells. Therefore, we focused screening on a compact collection of approximately 500 small-molecule compounds with a variety of different irreversible warheads. The screen was performed on a HEK293-based EGFP reporter cell line[46] that carries an out-of-frame EGFP gene, the expression of which can be restored upon CRISPR-Cas9 targeting (Fig. 1a). Inhibition of CRISPR-Cas9 cellular activity by inhibitors will lead to reduced fraction of activated EGFP cells (Supplementary Fig. 1). The first round of screening identified several Michael acceptor-bearing compounds that exhibited efficient inhibition of EGFP activation. The second round of screening was performed on a focused library of marketed or investigational compounds containing Michael acceptors (Fig. 1b). Several candidate CRISPR-Cas9 inhibitors were identified with half maximal inhibitory concentrations (IC$_{50}$) of less than 20 μM (Supplementary Table 1). Notably, an FDA-approved anticancer drug KPT330[47,48] displayed notable potency for Cas9 inactivation (Fig. 1b).

Inspired by the results with KPT330, we investigated other selective inhibitors of nuclear export (SINEs) including KPT185, KPT276, KPT335 and KPT8602 for Cas9 inhibition. It was found that these SINEs could efficiently suppress CRISPR-Cas9-induced EGFP activation (Fig. 1c and Supplementary Fig. 2a). To examine the inhibitory effects of KPT330 on CRISPR-Cas9 at the endogenous genomic sites, we designed four sgRNAs targeting to human *EMX1* (sg*EMX1-1* and sg*EMX1-2*), *AAVS1* and *HBB* genes, respectively. Similar to the procedure in EGFP activation assay, 10 μM KPT330 was added into the medium following the transfection of sgRNA- and Cas9-encoding plasmids in HEK293T cells. T7E1 assay analysis confirmed the efficient inhibition of CRISPR-Cas9 by KPT330 at endogenous genomic sites (Supplementary Fig. 2b). Similarly, other SINEs at 10 μM could inhibit CRISPR-Cas9 activity across different genomic sites (Supplementary Fig. 2c). Next-generation sequencing (NGS) analysis showed that treatment with SINEs at a therapeutic concentration of 0.5 μM[49] exhibited modest inhibition of CRISPR-Cas9 activity at *HBB* and *RUNX1* genes in HEK 293T cells (Fig. 1d). Thus, subsequent experiments with SINEs were performed at 0.5 μM concentration.

### SINEs indirectly inhibit Cas9 activity by interfering with the nuclear export of Cas9 mRNA. 
We next sought to explore the mechanism of action of SINEs for Cas9 inhibition. Surprisingly, it was found that KPT330 at a high concentration of 10 μM showed very limited inhibition of the activity of purified Cas9 protein in an in vitro DNA cleavage reaction (Supplementary Fig. 3a). This result suggested that KPT330 as well as other SINEs might inhibit CRISPR-Cas9 activity in an indirect manner. To test this hypothesis, we examined the effects of KPT330 on the cellular activity of Cas9-sgRNA ribonucleoproteins (RNPs). T7E1 analysis showed that 0.5 μM KPT330 exhibited very limited inhibition of Cas9-sgRNA RNP-mediated genome editing at *EMX1* and *AAVS1* sites in Hela cells (Supplementary Fig. 3b). Comparison of the effects of SINEs on transiently transfected plasmids (Fig. 1d) and on RNP of CRISPR-Cas9 (Supplementary Fig. 3a, b) indicated that SINEs were not direct inhibitors of CRISPR-Cas9 but rather functioned at transcriptional or translational level.

It has been well established that KPT330 exerts its anti-cancer activity by inhibiting nuclear transporter protein exportin-1 (XPO1), also known as CRM1[50]. XPO1 is responsible for the nuclear export of various tumor suppressor proteins and can, in association with adapter proteins, export a wide range of RNA cargos including messenger RNAs (mRNAs), ribosomal RNA

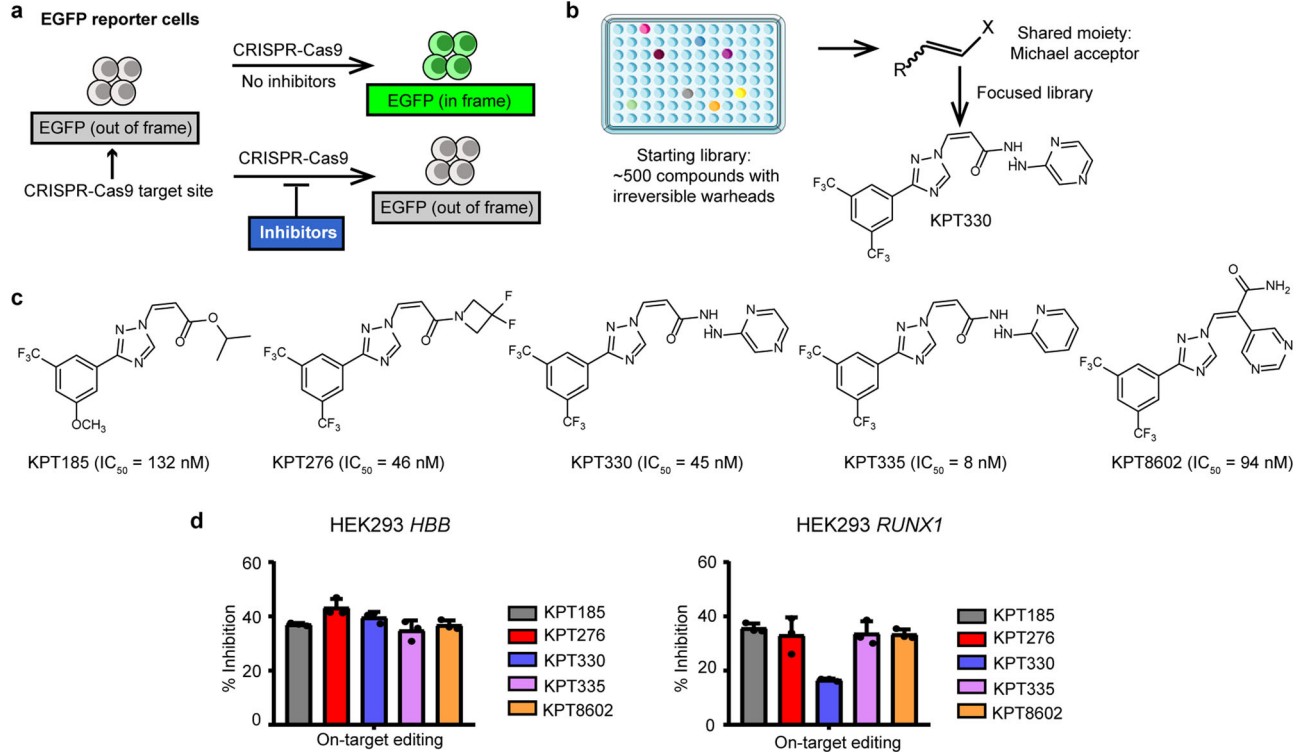

**Fig. 1 Identification of SINEs as CRISPR-Cas9 inhibitors. a** Schematic presentation of EGFP reporter cell-based screening of CRISPR-Cas9 inhibitors. **b** Flowchart of the screening process. X, electron withdrawing group. **c** Potency of SINEs on inhibiting CRISPR-activated EGFP fluorescence. **d** SINEs at 0.5 μM inhibit the on-target editing activity of CRISPR-Cas9 at *HBB and RUNX1* genes in HEK293 cells. The data are shown as mean ± SD ($n = 3$ biologically independent samples).

(rRNAs), transfer RNA (tRNAs), small nuclear RNAs (snRNA), microRNA (miRNA), and viral mRNAs[51]. To investigate whether SINEs could interfere with the nuclear export of Cas9 mRNA, we analyzed the effects of 0.5 μM KPT330 on the cytoplasmic expression of Cas9 mRNA following transfection of sgRNA-coding and Cas9-coding plasmids in HEK293T cells. We quantified the Cas9 mRNA in cytoplasmic fraction and total cell lysate in the KPT-treated and untreated groups. It was found that 0.5 μM KPT330 treatment reduced the cytoplasmic Cas9 mRNA fraction over the total fraction, independently of Cas9-associated sgRNA (Fig. 2a). These results suggested that KPT330 inhibited mRNA transport rather than transcription. Moreover, we evaluated the effects of other SINEs on the expression of cytoplasmic Cas9 mRNA and found that all SINEs reduced cytoplasmic Cas9 mRNA in a dose-dependent manner except KPT276 (Fig. 2b). To confirm SINEs affected Cas9 mRNA by interfering with the transport process rather than affecting the stability of cytoplasmic mRNA, we directly transfected cells with in vitro transcribed Cas9 mRNA along with sgRNA to bypass the nuclear export process associated with plasmid-encoded Cas9. It was found that 0.5 μM KPT330 did not significantly inhibit the genome modification events mediated by in vitro transcribed Cas9 mRNA (Supplementary Fig. 3c), indicating that KPT330 had little impact on Cas9 mRNA stability. These data collectively demonstrated that the inhibitory activity of SINEs was dependent on the nuclear export of Cas9 mRNA.

**XPO1 was involved in SINE-mediated inhibition of Cas9 mRNA nuclear export.** The cellular target of SINEs, XPO1, is an extensively characterized transporter protein for nuclear protein and RNA cargos. It is therefore possible that SINEs affect the nuclear export of Cas9 mRNA in an XPO1-dependent manner. In the presence of 0.5 μM KPT330, XPO1 knockdown by siRNA

(Fig. 2c, d) further reduced cytoplasmic Cas9 mRNA (Fig. 2e), suggesting that XPO1 was involved in SINE-mediated inhibition of Cas9 mRNA transport. It was noted that XPO1 siRNA alone did not impact Cas9 mRNA transport (Fig. 2e), likely due to the residual XPO1 (Fig. 2c, d) or alternative transport pathway.

It has been reported that XPO1 exports nuclear mRNA via adapter proteins human antigen R (HuR) or leucine-rich pentatricopeptide repeat protein (LRPPRC)[51]. RNA immunoprecipitation (RIP) experiments confirmed that HuR or LRPPRC could interact with Cas9 mRNA that was produced from Cas9-encoding plasmid both in the absence or presence of KPT330 (Fig. 2f). Importantly, KPT330 treatment reduced intracellular XPO1 as described[52] but not HuR or LRPPRC (Fig. 2g), suggesting that SINEs affected Cas9 mRNA transport by targeting XPO1 rather than adapter protein HuR or LRPPRC. Interestingly, KPT330 appeared to have opposite effects on XPO1 at protein and mRNA levels (Fig. 2d, g). It is known that KPT330 reaction with XPO1 can induce proteasome-dependent protein degradation, and the reduced intracellular XPO1 protein will in turn have a positive feedback on the expression of XPO1 mRNA[52–54]. Therefore, our results were consistent with the findings in previous studies. Collectively, these results have demonstrated that SINEs regulated the nuclear export of Cas9 mRNA via XPO1/HuR or XPO1/LRPPRC pathway (Fig. 2h).

**SINEs improve the specificity of Cas9-mediated genome editing in human cells.** Next, we sought to establish SINEs as modulatory compounds for controlling the cellular activity of CRISPR-Cas9 tools at the mRNA transport level. We analyzed the effects of 0.5 μM SINEs on the on- and off-target editing of CRISPR-Cas9 at *HBB* and *RUNX1* sites in human cells (Supplementary Fig. 4a). Importantly, it was found that SINEs improve

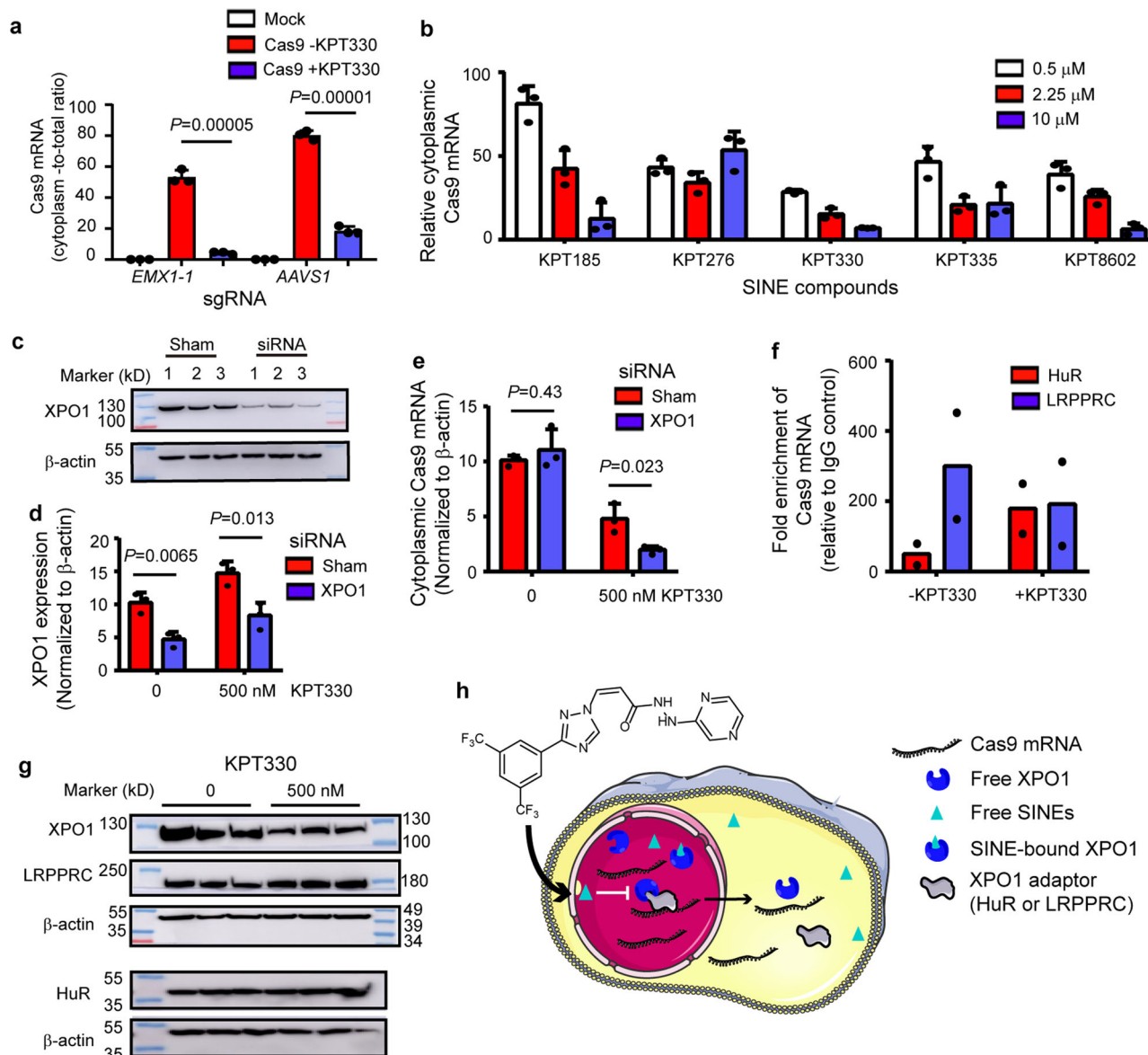

**Fig. 2 SINEs inhibit the nuclear export of Cas9 mRNA in an XPO1-dependent manner. a** KPT330 reduces cytoplasmic Cas9 mRNA, as determined by the ratio of cytoplasmic Cas9 mRNA to total Cas9 mRNA using RT-qPCR. Significant difference is determined using Student's *t*-test. **b** SINEs reduce cytoplasmic Cas9 mRNA in a dose-dependent manner, as determined by RT-qPCR. mRNA expression is normalized to drug-free group. **a**, **b** β-actin is used as an internal control. **c** Screening siRNA screen for XPO1 knockdown. **d** Evaluation of the efficiencies of siRNA knockdown of XPO1 mRNA expression in the absence and presence of 0.5 μM KPT330, as determined by RT-qPCR. **e** The effects of XPO1 knockdown on Cas9 mRNA transport in the absence and presence of 0.5 μM KPT330. **d**, **e** Significant difference between sham and XPO1 knockdown is determined using Student's *t*-test. **f** RIP experiment showing HuR and LRPPRC binding with Cas9 mRNA. **g** The effects of KPT330 treatment on the expression of XPO1, HuR and LRPPRC proteins. **h** Cartoon illustrating SINE-mediated modulation of Cas9 mRNA transport. The above data are shown as mean ± SD ($n = 2$ or 3 biologically independent samples).

the specificity of CRISPR-Cas9, as defined by the ratio between on-target and off-target activity (Fig. 3a).

As previous studies have shown that Acrs can improve the specificity of CRISPR-Cas9[37], we intended to directly compare SINEs with Acrs for their effects on CRISPR-Cas9 targeting. It appeared that KPT330 and KPT8602 had greatest improvement on CRISPR-Cas9 specificity at *HBB* and *RUNX1* sites, respectively. Therefore, we focused subsequent analysis on these two SINE compounds. Meanwhile, strong CRISPR-Cas9 inhibitor AcrIIA4[38] and weak inhibitors AcrIIC1[32] or AcrIIC3[32] were selected as comparison. The comparative experiments were performed across different cell types and genomic sites. We found that KPT330 and KPT8602 treatment could consistently improve the specificity of CRISPR-Cas9 by preferentially

inhibiting the off-target activity. Importantly, under all treatment conditions with KPT330 and KPT8602 a considerable fraction of the on-target activity of CRISPR-Cas9 was retained (Fig. 3b–f). By contrast, neither strong nor weak Acr inhibitors could consistently improve the specificity of CRISPR-Cas9 (Fig. 3b–f). Although AcrIIA4 appeared to exhibited higher specificity than SINEs at *EMX1-3* site (Fig. 3f), this benefit was associated with markedly compromised on-target activity (Fig. 3f and Supplementary Fig. 4b) that prevented the practical applications of AcrIIA4 for modulating CRISPR-Cas9 activity and specificity. Although AcrIIA4 appeared to exhibited higher specificity than SINEs at *EMX1-3* site (Fig. 3f), this benefit was associated with markedly compromised on-target activity (Fig. 3f and Supplementary Fig. 4b) that prevented the practical applications of

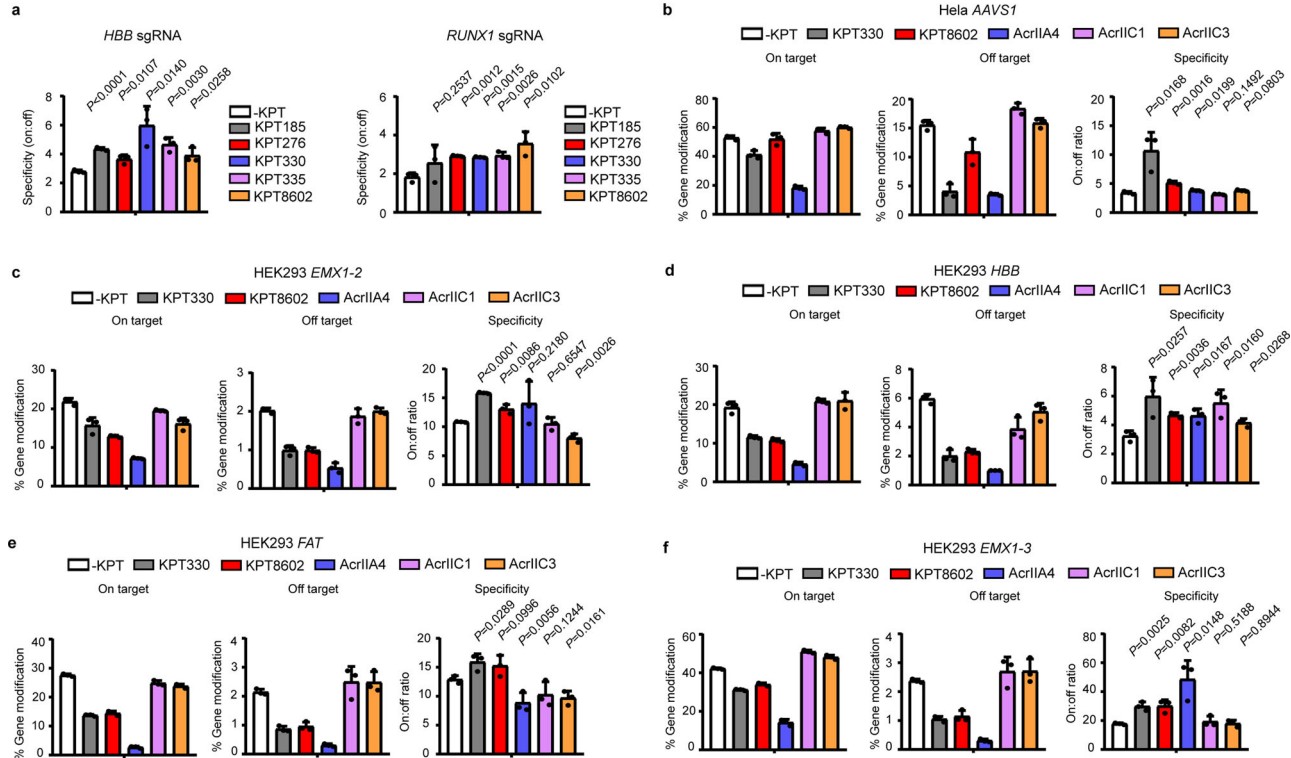

**Fig. 3 Evaluation of the effects of SINEs on the genome-editing specificity of CRISPR-Cas9. a** SINEs at 0.5 μM improve the genome-editing specificity of CRISPR-Cas9 at *HBB* gene and *RUNX1* in HEK293 cells. **b** Comparison of the effects of SINEs and Acrs on the genome-editing activity of CRISPR-Cas9 at *AAVS1* site in Hela cells. **c**–**f** Comparison of the effects of SINEs and Acrs on the genome-editing activity of CRISPR-Cas9 at *EMX1-2*, *HBB*, *FAT*, and *EMX1-3* in HEK293 cells. The data are shown as mean ± SD (*n* = 3 biologically independent samples) and the significant difference between mock and SINE or Acrs treatment is determined using Student's *t*-test.

AcrIIA4 for modulating CRISPR-Cas9 activity and specificity. In addition to Acrs, high-fidelity Cas9 (HF-Cas9) bearing mutations N497A/R661A/Q695A/Q926A has been developed and been reported to bear dramatically reduced off-target activity[55]. Although HF-Cas9 had restricted off-target activity compared with wild-type Cas9, its on-target activity was largely reduced at examined the genomic sites (Supplementary Fig. 4c).

Importantly, KPT330 treatment did not alter the pattern of Cas9-induced mutations, as characterized by the mutation peak upstream of the protospacer adjacent motif (PAM) sequence (Supplementary Fig. 5a). In addition, KPT330 had minor impact on the length (Supplementary Fig. 5b) or frame phase of CRISPR-Cas9-induced indels (Supplementary Fig. 5c). These results suggested that SINEs could modulate the genome-editing activity and specificity of CRISPR-Cas9 without complicating the genomic outcome. It must be noted, however, that the effects of SINEs on CRISPR-Cas9 were dependent on genomic sites, sgRNAs or compounds.

In addition to SINEs, we investigated the effects of leptomycin B, an anti-fungal antibiotics known to inhibit XPO1[56], on the genome-editing activity of CRISPR-Cas9. It was found that 500 nM leptomycin B could efficiently inhibit the on- and off-target editing of CRISPR-Cas9 at the *HBB* site in human cells (Supplementary Fig. 6a). Under an adjusted concentration of 100 nM, leptomycin B exhibited more inhibition toward the off-target editing activity and thus improved the specificity of CRISPR-Cas9 (Supplementary Fig. 5b). These results supported the above observations that targeting XPO1 could be a feasible approach to modulating CRISPR-Cas9 activity and specificity.

**SINEs improve the editing window of CBE.** We next investigated the effects of SINEs on Cas9-derived base editing tools. It

has been reported that limiting the exposure of cells to excessive CBEs could improve their editing specificity[13,17]. We hypothesized that the wide targeting window of CBEs is attributed, at least in part, to the uncontrolled activity of base editing agents inside cells, which can be improved by temporally controlling the cellular activity of SINEs. It was found that 0.5 μM KPT330 could inhibit BE3 CBE (rAPOBEC1-nCas9-UGI)-induced C-to-T conversion at various genomic sites (Supplementary Fig. 7a). Similar inhibitory activities of other SINEs were observed with BE3 (Supplementary Fig. 7b) and A3A (hAPOBEC1-nCas9-UGI) (Supplementary Fig. 7c) CBEs. Importantly, KPT185, KPT330, KPT335, and KPT8602 exhibited more inhibition toward out-of-window editing than on-target editing of BE3 or A3A CBEs (Fig. 4a).

The preferential inhibitory activity of SINEs at out-of-window positions allowed a degree of control over the targeting scope of CBEs. To explore the potential application of SINEs during CBE-mediated gene correction, we assessed the effects of KPT330 on A3A CBE editing in a cell-based disease model of Marfan syndrome, where a T7498C mutation was introduced into the *FBN1* gene of HEK293T cells to model the pathogenic C2500R mutation[57]. It was found that KPT330 at 0.5 μM inhibited both on-target and out-of-window editing (Fig. 4b) of A3A CBE but a 2-fold selectivity of inhibition toward out-of-window over on-target editing was observed (Fig. 4b).

The above results suggested that SINEs could inhibit different CRISPR tools carrying the Cas9 module, thus we investigated the effects of SINEs on the recently developed prime editing (PE) tool[58]. Following transfection of PEs, 0.5 μM SINEs were supplemented to cell culture and their effects on PE-induced insertions, deletions and point mutations were analyzed by NGS. It was found that all SINEs showed inhibitory activity toward PEs

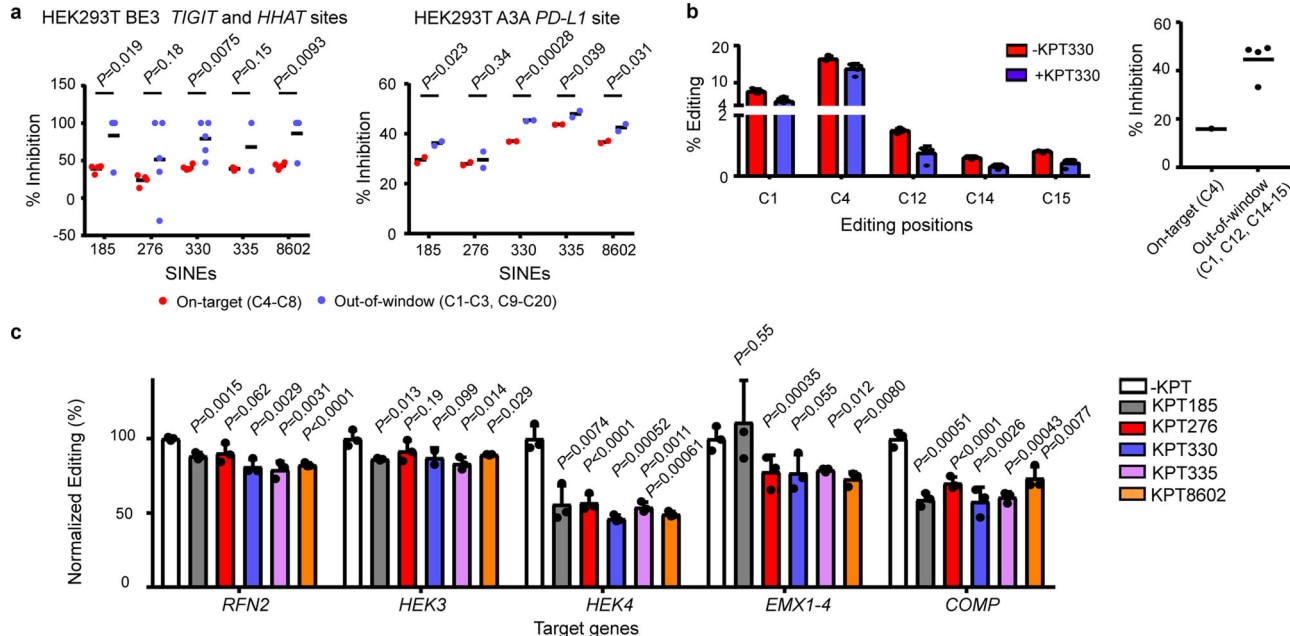

**Fig. 4 Evaluation of the effects of SINEs on base- and prime- editing tools. a** The effects of 0.5 μM SINEs on the on-target and out-of-window editing of BE3 and A3A CBEs at different genomic sites. **b** KPT330 at 0.5 μM preferentially inhibits the out-of-window over on-target editing of A3A CBE at the pathogenic *FBN1* site in a HEK293-based disease model of Marfan syndrome. **c** SINEs inhibit the editing efficiency of PE2 and PE3 in HEK293T cells. The data are shown as mean ± SD (*n* = 3 biologically independent samples) and significant difference between mock and SINE treatment is determined using Student's *t*-test.

across different genomic sites in HEK293T cells. The degree of inhibition seemed to be dependent on both the type of SINEs and the edited sites (Fig. 4c). These results again suggested that SINEs functioned as general inhibitors to CRISPR-Cas9, independent of its fusion partners.

## Discussion

Under therapeutic settings, CRISPR-based genome engineering tools are often delivered into human cells as episomal DNA by adeno-associated viruses (AAVs)[6]. In these applications, CRISPR-Cas9 is associated with off-target effects that are arised from the constitutively active Cas9 protein[7–9]. To overcome these problems, temporal and spatial control strategies have developed to modulate the intracellular activity of CRISPR-Cas9[8,59–62]. For example, inducible promoters have been devised to regulate CRISPR-Cas9 activity at the transcriptional level[63]. Fusion of small-molecule responsive elements to Cas9 proteins allowed the control of CRISPR-Cas9 activity at the post-translational level[60]. In addition, the specificity of CRISPR-Cas9 targeting can be improved by restricting the intracellular half life of Cas9 protein using directly delivered Cas9-sgRNA RNP[18]. These experiments have demonstrated that the intracellular activity of CRISPR-Cas9 can be regulated at multiple levels during the life cycle of CRISPR-Cas9 inside cells.

CRISPR-Cas9 inhibitors provide compelling opportunity to prevent the excessive intracellular activity of CRISPR-Cas9. Despite of the distinct mechanisms of action, most existing CRISPR-Cas9 inhibitors directly intervene the targeting or cleavage processes of CRISPR-Cas9, thus known as direct inhibitors. Specifically, a previous study has established a high-throughput platform to screen for small molecules that can inhibit the in vitro cleavage activity of Cas9[45]. Different from this in vitro screening system, the present study relied on an EGFP reporter cell-based assay to screen for small molecules that could exert Cas9-inhibiting activity under intracellular conditions. Surprisingly, the best hit of compounds turned out to be an indirect inhibitor that

disrupted the nuclear export process of Cas9 mRNA. This compound and its analogs are known as selective inhibitors of nuclear export (SINEs), which are developed to target the nuclear transporter protein XPO1 for anti-cancer treatment. Mechanistic study have revealed that SINE-mediated CRISPR-Cas9 inhibition relies on XPO1 and its adapter proteins HuR and LRPPRC, which are required for mRNA transport. The consistent mechanism of action renders SINEs optimum choice for repurposing for modulating CRISPR-Cas9 activity. Importantly, one of the identified SINE compounds, KPT330, has been approved to treat relapsed or refractory multiple myeloma (RRMM)[47] and relapsed or refractory diffuse large B-cell lymphoma (DLBCL)[48]. To the best of our knowledge, KPT330 is thus far the only FDA-approved small molecule that has been identified to bear CRISPR-Cas9-modulating activity. More interestingly, KPT330 as well as other SINEs may improve the genome-editing specificity of CRISPR-Cas9 and the targeting scope of Cas9-derived CBEs. Although these improvements are relatively modest, the comparative analysis has demonstrated that SINEs outperformed traditional Acrs in improving the specificity of CRISPR-Cas9 across different cell types and genomic sites. Furthermore, our discovery provides an alternative approach of manipulating the activity of CRISPR-Cas9 at the mRNA transport level and shed the light on developing next-generation compounds targeting the nuclear export of Cas9 mRNA. In addition, because HuR and LRPPRC are also involved in XPO1-dependent Cas9 mRNA nuclear export, it is possible that the intracellular activity of CRISPR-Cas9 can be regulated by targeting HuR and LRPPRC. While our study has suggested the feasibility of regulating Cas9 mRNA transport using SINEs, it must be noted that these compounds are designated to target XPO1 for inhibiting the mRNA transport of oncogene[64]. Therefore, the global cellular activity of SINEs on the transport of mRNA or other types of RNA should be considered during their applications with CRISPR.

One interesting discovery in this study was that the knockdown of XPO1 alone did not affect the content of Cas9 mRNA in the

cytoplasm. This could be explained by the function of residual XPO1 or the existence of alternative transport pathway of Cas9 mRNA. It is known that in addition to XPO1, NXF1 protein can mediate the nuclear export of mRNA in mammalian cells[50]. It would be thus interesting to explore the function of NXF1 during Cas9 mRNA export in future studies.

Previously established approaches to modulating CRISPR-Cas9 activity and specificity include anti-CRISPR proteins[37] or peptide[31]. Although these methods can improve the specificity of CRISPR-Cas9, they require transient transfection of polypeptide-coding plasmids, which may limit the practical applications. In addition, these protein or peptide-based CRISPR inhibitors may be associated with considerable loss of the on-target activity of CRISPR-Cas9. Small-molecule CRISPR-Cas9 inhibitors have been developed and employed to modulate the activity of CRISPR-Cas9-based genome-editing and base-editing tools[45]. Besides these inhibitory molecules, high-fidelity Cas9 variants have been developed and been shown to be capable of reducing the off-target activity. However, these HF-Cas9 may have compromised on-target activity. Therefore, the discovery of SINEs as indirect Cas9 inhibitors presents an alternative approach to modulating CRISPR-Cas9 activity and specificity.

In summary, we report in the present study the discovery of SINEs as safe-in-human, indirect inhibitors for CRISPR-based genome editing, base editing and prime editing tools. Our study highlights the importance of Cas9 mRNA transport to the activity and specificity of CRISPR-Cas9. The ability to manipulate Cas9 mRNA transport by SINEs promises a degree of control over the genome editing activity of CRISPR-Cas9. Our discovery should speed advances toward the practical applications of CRISPR-Cas9 technology.

## Methods

**Cell culture**. HEK293T, HEK293-based EGFP reporter and Hela cells were maintained in Dulbecco's Modified Eagle's medium (DMEM, ThermoFisher Scientific) supplemented with 10% fetal bovine serum (FBS, ThermoFisher), 100 IU/mL of penicillin and 100 μg/mL of streptomycin at 5% $CO_2$ and 37 °C in a fully humidified incubator and were passaged when 70–90% confluence was reached. Cells were cultured in RPMI-1640 medium supplemented with 10% FBS, at 37 °C under 5% $CO_2$. Hela cells were obtained from American Type Culture Collection (ATCC). HEK293T cells were obtained from the Cell Bank of Shanghai Institutes for Biological Science (SIBS). HEK293-based EGFP reporter cell line was constructed in a previous study[65]. Cell lines were validated by VivaCell Biosciences (Shanghai, China).

**EGFP reporter-based screen for CRISPR inhibitor**. HEK293-based EGFP reporter cells ($2.5 \times 10^4$ cells/well) were seeded on to 96-well plate. At 24 h after seeding, cells were transfected with 50 ng pST-Cas9 and 50 ng pU6-EGFP T2 plasmids using Opti-MEM medium (Gibco) and 0.2 μL lipofectamine 2000 (Invitrogen). At 6 h after transfection, Opti-MEM was replaced with DMEM supplemented with 10% FBS and compounds of indicated concentration. Positive control contains 50 ng pST-Cas9 and 50 ng pU6-EGFP T2 plasmids without compounds. Negative control contains 50 ng pST-Cas9. At 30 h after drug treatment, the fluorescence of cells was quantified by flow cytometry. A cut-off of 80% reduction in the percentage of cells with activated fluorescence was used for the screening experiments.

**Expression and purification of SpyCas9 proteins**. pET28b plasmids coding SpyCas9 WT proteins was transformed into *E. coli* BL21 (DE3) cells. Single colonies were picked and grown in 2 L LB media supplemented with 50 μg/mL kanamycin. Culture was grown to an $OD_{600}$ of 0.8. Protein expression was induced with 0.2 mM isopropyl-β-D-thiogalactopyranoside (IPTG) at 16 °C overnight. Cells from 2 L culture were pelleted by centrifugation at $6000 \times g$ at 4 °C for 15 min and then re-suspended in 40 mL binding buffer containing 20 mM TrisHCl, pH 8.0 and 0.5 M NaCl. Cell suspension was then supplemented with 1 mM Tris (2-carboxyethyl) phosphine (TCEP) and 1× complete inhibitor cocktail (Roche). Cells were lysed by sonication on ice and then centrifuged at $80,000 \times g$ at 4 °C for 30 min. The supernatant of cell lysate was incubated with 1 mL Ni-NTA agarose beads (QIAGEN) at 4 °C for 1 h. The resin was washed with 20 mL wash buffer that was made by supplementing binding buffer with 30 mM imidazole. Proteins were eluted with 5 mL elute buffer that was made by supplementing binding buffer with 300 mM imidazole. Eluted protein samples were further purified by gel filtration using Superose 6 10/300 column (GE Healthcare). These proteins were buffer-

exchanged to storage buffer containing 20 mM HEPES, pH 8.0 and 200 mM NaCl, aliquoted and stored at −80 °C.

**In vitro transcription of sgRNA**. sgRNA was transcribed from a 100 ng sgRNA-coding PCR product with a 5′ T7 promoter sequence using HiScibe T7 Quick High yield RNA Synthesis kit (NEB). The transcription was performed at 37 °C overnight and then purified by phenol: chloroform extraction, followed by ethanol precipitation. Purified sgRNA was quantified by spectrometry and stored at −80 °C.

**In vitro DNA cleavage**. In vitro cleavage assay was performed as described[46]. Cas9 protein and transcribed sgRNA were incubated for 10 min at room temperature in reaction buffer containing 1× NEB buffer 3.1 (NEB Biolabs) supplemented with 1 mM DTT to form Cas9-sgRNA RNP complex. Cleavage was performed in 10 μL reactions containing 100 ng of substrate DNA and 1 μL RNP complex of indicated concentrations at room temperature for 1 h. Reactions were terminated by addition of 1× DNA loading buffer and resolved on 2% agarose gels. For inhibition experiments, KPT330 were dissolved in reaction buffer and incubated with Cas9 protein or pre-assembled Cas9-sgRNA RNP for 10 min at room temperature and the mixed solution was then added to the in vitro cleavage reaction.

**RT-qPCR quantification of Cas9 mRNA**. HEK293 cells ($5 \times 10^5$ cells/well) were seeded on to 6 cm plate. At 24 h after seeding, cells were transfected with 2 μg pST-Cas9 plasmids and 1 μg of either pU6-EMX1 T2 or pU6-AAVS1 T2 (Supplmenetary Table 2) using Opti-MEM medium and 6 μL lipofectamine 2000. At 6 h after transfection, Opti-MEM was replaced with DMEM supplemented with 10% FBS in the absence or presence of 10 μM KPT330.

At 36 h after transfection, cells were harvested and the cytoplasmic and nuclear fractions were separated using a commercial kit (ThermoFisher, AM1921). Briefly, collected cells were washed with PBS and resuepended in cell fraction buffer, incubated on ice for 5 min and centrifuged at $500 \times g$ at 4 °C for 5 min. The cytoplasmic fraction was carefully extracted from the nuclear pellet. The nuclear pellet was washed with ice-cold cell fractionation buffer and lysed with cell disruption buffer. The separated cytoplasmic or nuclear fraction was mixed with an equal volume of 2× lysis/binding solution and then one sample volume of 100% ethanol was added. The ethanol-treated samples were run through a filter cartridge, washed once with 700 μL wash solution 1, twice with 500 μL wash solution 2 and 3. The RNA samples were eluted with 40–60 μL of pre-heated 95 °C elution solution and then eluted a second time with 10–60 μL of elution solution. RNA of each sample was reverse transcribed into cDNA using PrimeScript RT reagent Kit with gDNA Eraser (Takara, RR047A). The cDNA was amplified using a SYBR Green Master Mix Kit (Takara, RR420A) on Applied Biosystems Q6 Real-Time PCR cycler. The following thermal conditions were used for Cas9, XPO1 and Actin: initial denaturation at 95 °C for 30 s, followed by 40 cycles of 95 °C for 5 s, 60 °C for 30 s, and 95 °C for 15 s, with a final extension at 60 °C for 1 min followed by 95 °C for 15 s.

**The effects of SINEs on genome-editing activities of directly delivered Cas9 mRNA**. HEK293T cells ($2.5 \times 10^5$ cells/well) were seeded on to 24-well plate. At 24 h after seeding, cells were transfected with 3 μg in vitro transcribed Cas9 mRNA (APExBIO, R1014) and 2 μg sgRNA using Opti-MEM medium and 6 μL lipofectamine 2000. At 6 h after transfection, Opti-MEM was replaced with DMEM supplemented with 10% FBS and 0.5 μM KPT330 is supplemented to cell culture immediately. To analyze genome-editing outcome, cells were harvested at 36 h after transfection and the genomic DNA was extracted using QuickExtract DNA Extraction Solution (Epicenter). Genomic PCR reaction was performed using 100 ng genomic DNA, corresponding primers (Supplementary Table 5), Phanta Max Super-fidelity DNA Polymerase (Vazyme) using a touchdown cycling protocol (30 cycles of 98 °C for 10 s, 68–58 °C for 15 s, and 68 °C for 60 s). The PCR products were digested by T7E1 enzyme (NEB), resolved on 2% agarose gel and then analyzed by densitometry measurements as described[66]. Three biological replicates were performed for each condition.

**The effects of KPT330 on the genome-editing activity of directly delivered Cas9-sgRNA RNP**. Hela cells ($2 \times 10^5$ cells/well) were harvested, washed with PBS and re-suspended in 20 μL of SE nucleofection buffer (Lonza). Cas9 protein (10 μg) and 15 μg transcribed sgRNA were incubated for 10 min at room temperature to form Cas9-sgRNA RNP complex. Hela cells and Cas9-sgRNA RNP was nucleofected by Lonza 4D nucleofector. Immediately following the nucleofection, 100 μL pre-warmed medium was added into nucleofection cuvettes and the cells were transferred to culture dishes. At 2 h after nucleofection, the cells were attached to the culture dish and 0.5 μM KPT330 was added into cell culture. At 36 h of post nucleofection, cells were harvested and genomic DNA was extracted, PCR amplified and analysed by T7E1 assay[66]. Three biological replicates were performed for each condition.

**The effects of SINEs and Acrs on CRISPR-Cas9 activity in human cells**. HEK293T, HEK293-based EGFP reporter and Hela cells ($2 \times 10^5$ cells/well) were

seeded on to 24-well plate. At 24 h after seeding, cells were transfected with 500 ng pST-Cas9 plasmids and 500 ng empty vector pcDNA3.1, AcrIIA4-pcDNA3.1, AcrIIC1-pcDNA3.1, or AcrIIC3-pcDNA3.1 plasmid and 250 ng of either pU6-AAVS1 T2, pU6-HBB T2, pU6-VEGFA T2, pU6-IL2RG T2, pU6-RUNX1 T2 or pU6-EMX1 T2, pU6-FAT using Opti-MEM medium and 2.5 μL lipofectamine 2000. At 6 h after transfection, Opti-MEM was replaced with DMEM supplemented with 10% FBS in the presence or absence of 0.5 μM SINEs of indicated concentrations. Cells were harvested at 36 h after transfection and the genomic DNA was extracted. The genomic DNA of sorted cells was extracted and PCR amplified as described above for next-generation sequencing analysis. Three biological replicates were performed for each condition.

**The effects of leptomycin B on CRISPR-Cas9 activity in human cells.** HEK293T ($2 \times 10^5$ cells/well) were seeded on to 24-well plate. At 24 h after seeding, cells were transfected with 500 ng pST-Cas9 plasmids and 250 ng pU6-HBB T2 or pU6-EMX1 T2 plasmids using Opti-MEM medium and 1.5 μL lipofectamine 2000 (Invitrogen). At 6 h after transfection, Opti-MEM was replaced with DMEM supplemented with 10% FBS and leptomycin B at indicated concentrations (100 or 500 nM). At 36 h after transfection, the genomic DNA of treated cells was extracted and PCR amplified as described above for next-generation sequencing analysis. Three biological replicates were performed for each condition.

**The effects of SINEs on high fidelity CRISPR-Cas9 activities in human cells.** HEK293T ($2 \times 10^5$ cells/well) were seeded on to 24-well plate. At 24 h after seeding, cells were transfected with 500 ng HF-Cas9 plasmids (N497A/R661A/Q695A/Q926A) and 250 ng of either pU6-AAVS1 T2, pU6-HBB T2 or pU6-EMX1 T2 plasmids using Opti-MEM medium and 1.5 μL lipofectamine 2000 (Invitrogen). At 6 h after transfection, Opti-MEM was replaced with DMEM supplemented with 10% FBS in the presence or absence of 0.5 μM KPT330. At 36 h after transfection, the genomic DNA of treated cells was extracted and PCR amplified as described above for next-generation sequencing analysis. Three biological replicates were performed for each condition.

**The effects of SINEs on CRISPR-based BE3 and A3A CBE activities in human cells.** HEK293T ($2 \times 10^5$ cells/well) were seeded on to 24-well plate. At 24 h after seeding, cells were transfected with 500 ng CBE-encoding plasmid and 250 ng sgRNA plasmids using Opti-MEM medium. At 6 h after transfection, Opti-MEM was replaced with DMEM supplemented with 10% FBS in the absence or presence of 0.5 μM SINEs. At 36 h after transfection, genomic DNA was extracted and then analysed by NGS as described above. Three biological replicates were performed for each condition.

**The effects of SINEs on CRISPR-based prime editing in human cells.** HEK293T ($1 \times 10^5$ cells/well) were seeded on to 48-well plate. At 24 h after seeding, cells were transfected with 750 ng PE2 plasmid, 250 ng pegRNA plasmids and 83 ng nicking sgRNA (Supplementary Table 4) using Opti-MEM medium. At 6 h after transfection, Opti-MEM was replaced with DMEM supplemented with 10% FBS in the absence or presence of 0.5 μM SINEs. At 72 h after transfection, genomic DNA was extracted, The genomic DNA of sorted cells was extracted and PCR amplied as described above. Three biological replicates were performed for each condition.

**RNAi experiments.** HEK293T ($2 \times 10^5$ cells/well) were seeded on to 12-well plates. At 24 h after seeding, the cells in each well were transfected with 20 pmol siRNA targeting to human XPO1 (GenePharma, Shanghai, China) (Supplementary Table 3) using lipofectamine 2000 (Invitrogen). At 6 h after transfection, Opti-MEM was replaced with DMEM supplemented with 10% FBS. At 24 h after siRNA transfection, cells were transfected with 1 μg pST-Cas9 plasmids and 0.5 μg pU6-AAVS1 T2 plasmids using lipofectamine 2000 in Opti-MEM medium. At 30 h after siRNA transfection, Opti-MEM was replaced with DMEM supplemented with 10% FBS in the absence or presence of 0.5 μM KPT330. At 30 h after KPT330 treatment, cells were harvested and RNA was fractionated and extracted using RNA extraction kit (ThermoFisher). The cytoplasmic RNA was reverse transcribed into cDNA (Takara) and then amplified using a SYBR Green Master Mix Kit (Takara) in real-time PCR detection system.

**Western blot.** Cells were harvested by centrifugation, washed once with PBS and lysed in RIPA buffer (Life Technologies) according to the manufacturer's instructions. Protein concentrations of cell lysates were determined by bicinchoninic acid (BCA) method according to the manufacturer's recommendations (Beyotime Biotechnology). For each sample, 15 μg total protein was loaded and resolved on Bis–Tris 4–12% gels (GenScript) and transferred to a polyvinylidene difluoride (PVDF) membrane using Tris-Glycine Buffer (Sangon Biotech). XPO1, HuR, LRPPRC, and β-actin were detected using rabbit anti-XPO1 monoclonal antibody (mAb; Cell Signaling, 46249S, 1:1000), anti-HuR mAb (Abcam, ab200342, 1:1000) and anti-LRPPRC mAb (Abcam, ab205022, 1:1000) and mouse anti-β-actin mAb with HRP conjugate (Cell Signaling, 12262S, 1:1000). HRP-linked anti-rabbit IgG antibody (Cell Signaling, 7074P2, 1:5000) was used as the secondary antibody. Meilunbio fg super sensitive ECL luminescence reagent (Meilunbio). The uncropped images are provided in Supplementary Fig. 8.

**RNA immunoprecipitation (RIP).** HEK293T cells were seeded on to 15 cm dish. At 24 h after seeding, cells were transfected with 40 μg pST1374-Cas9 plasmid at 80% confluence using lipofectamine 2000. At 6 h after transfection, Opti-MEM was replaced with DMEM supplemented with 10% FBS in the absence or presence of 0.5 μM KPT330. At 36 h after transfection, the medium was removed and cells were washed twice with ice-cold PBS and then harvested. RIP experiments were performed Imprint RNA Immunoprecipitation Kit (Sigma-Aldrich, RIP-12RXN). Following manufacturer's instructions, 3.75 μg LRPPRC (Abcam, ab205022) or 3.75 μg HuR (Abcam, ab200342) antibodies was added into each RIP reaction. IgG (Sigma-Aldrich, M7023) of 3.75 μg was used as a reference. RT-qPCR was used to quantify immunoprecipitation-purified Cas9 mRNA using IgG-enriched mRNA as a reference for non-specific enrichment. The difference of RNA sample preparation was accounted by normalizing the Ct values of antibody-enriched RNA to that of the input RNA in the same RT-qPCR assay. Procedure of calculation is detailed in manufacturer's instructions. Western blot was performed to rule out the interference of KPT330 treatment on the protein expression of XPO1, HuR, and LRPPRC.

**Next-generation sequencing of edited genomic sites.** Edited genomic site was amplified by two consecutive PCR reactions. The PCR reaction was performed in a 50 μL reaction containing 200 ng genomic DNA, 0.5 μM forward and reverse primers with bridging sequences 5′-ggagtgagtacggtgtgc-3′ and 5′-gagttggatgctg-gatgg-3′ added to 5′ end of the target-binding sequences and Phanta Max Super-fidelity DNA Polymerase (Vazyme, Nanjing, China). The PCR cycling conditions were as follows: 95 °C for 3 min, 30 cycles of 95 °C for 15 s and 68 °C to 58 °C for 50 s and final extension at 68 °C for 2 min. The libraries were sequenced using Illumina HiSeq platform (Novogene Bioinformatics Institute, Beijing, China). The concentration of the libraries was determined using Qubit 2.0 (Life Technologies). The libraries were diluted to 1 ng/μL and the insert size of the libraries was analyzed by Agilent Bioanalyzer 2100 (Agilent). The SYBR green qRT-PCR protocol was used to accurately dose the effective concentration of the libraries. Three biological replicates were processed by Personabio (Shanghai, China) or Hi-TOM (Hangzhou, Zhejiang, China) using Illumina HiSeq NovaSeq or Hiseq X Ten platform.

Paired end reads with 150 bp length were selected and cleaned to remove adapter sequences and low quality paired reads. The following criteria were used to remove the low quality reads: (i) containing more than 10% "N" s; (ii) containing more than 50% bases having low quality values (Phred score less than 5); (iii) duplicated reads. The coverage values were calculated using SAMTtools[67].

For analysis of CRISPR-Cas9-meidated genome editing, amplicons with less than 6 M read counts were excluded from the analyses. Short reads were aligned to the reference sequence by Bowtie2[68] with the following parameters: -D 5 -R 3 -N 1 --gbar 1 --rdg 5,1 --rfg 5,1 --dovetail. Aligned reads were sorted by SAMtools[67] and indel calling was performed by mpileup[69] with maximum read depth per sample equal to the total reads mapped. VarScan v2.4[70] was used for the quality control of indels in mpileup output with a minimum variant frequency of ≥0.001, and a P value threshold of ≤0.05. With the above settings, the following items were quantified including the proportions of reads with Indels at each position in the 20 bp target region, the proportions of Indel with different insertion or deletion length, the proportions of Indel reading frames ($3N$, $3N+1$, and $3N+2$) and the proportions of reads harboring variants over the total number of aligned reads. For identification of Single Nucleotide Polymorphism (SNP), Bam-readcount was applied to report the numbers of all types of nucleotide ("A", "T", "G", "C", "N") at each position in the target region. The proportions of reads harboring mutations at each position were calculated by dividing over the total number of aligned reads at that position.

For analysis of base editing (BE) and prime editing (PE)[58] results, amplicon sequences were aligned to a reference sequence using CRISPResso2. Prime editing efficiency in percentage was calculated as:

$$[\# \text{ of reads with the desired edit that do not contain indels}] \div [\# \text{ of total reads}]$$
(1)

For analysis of point mutation, CRISPResso2 was run in standard mode with "discard_indel_reads" on. The frequency of PE-induced point mutations in percentage was then calculated as:

$$[\text{frequency of specified point mutation in non}-\text{discarded reads}]$$
$$\text{x} [\# \text{ of non}-\text{discarded reads}] \div [\text{total reads}]$$
(2)

For insertion or deletion edits, CRISPResso2 was run in HDR mode using the desired alleles as the expected alleles (with e flag) and with "discard_indel_reads" on. Editing yield was calculated as:

$$[\# \text{ of HDR aligned reads}] \div [\text{total reads}]$$
(3)

For all experiments, indel yields were calculated as:

$$[\# \text{ of indel}-\text{containing reads}] \div [\text{total reads}].$$
(4)

The processed data for the editing efficiency has been summarized in Supplementary Data 1.

**Statistics and reproducibility**. Two or three biological replicates were performed for each experimental condition. Significant difference was analyzed using two-tailed student's *t*-test.

**Reporting summary**. Further information on research design is available in the Nature Research Reporting Summary linked to this article.

## Data availability
NGS data have been deposited into NCBI Sequence Read Archive (SRA) database with the accession number PRJNA565327[71]. The raw data for the results presented in this study can be found in Supplementary Data 1. Uncropped blots and gels are presented in Supplementary Figs. 8 and 9. All other data are available from the corresponding authors on reasonable request.

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

## Acknowledgements

J.L. acknowledges the National Natural Science Foundation of China (NSFC, 31600686) and ShanghaiTech University Startup Fund (2019F0301-000-01) for their financial support. J.D. acknowledges NSFC 21672240, NSFC 21421002, Strategic Priority Research Program of the Chinese Academy of Sciences (XDB200203), the Key Research Program of Frontier Sciences from Chinese Academy of Science (QYZDB-SSWSLH-028) and Shanghai Sciences and Technology Committee (18JC1415500 and 18401933502) for their financial support. We thank Prof. Karl Barry Sharpless at Scripps Research and Prof. Kuilin Ding at Shanghai Institute of Organic Chemistry for revision of the manuscript and critical comments. We thank the High-Throughput Screening Platform and at Shanghai Institute for Advanced Immunochemical Studies (SIAIS) at ShanghaiTech University for the support of flow cytometry experiments. We thank Dr. Lichun Jiang and Biomedical Big Data Platform at SIAIS for the support of NGS analyses.

## Author contributions

J.L., J.D., and B.J. conceptualized study. J.L., J.D., S.J.W., and Y.R.C. designed the experiments and analyzed data. T.M., T.G., and G.M. collected and synthesized the compounds in this study. S.-J.W. performed the screen experiments. Y.R.C., S.J.W., and P.Y.H. performed the in vitro DNA cleavage assay and CRISPR, BE, PE and RIP experiments in human cells. J.C. analyzed NGS data. J.L. and J.D. wrote the manuscript. All authors discussed the results and commented on and approved the manuscript.

## Competing interests

The authors declare no competing interests.
