## [Peer Review File · Communications Biology]

Reviewers' comments:

Reviewer #1 (Remarks to the Author):

Cui, Wang et al. developed a well-designed assay to screen for covalent CRISPR-Cas9 inhibitors and made the novel discovery that selective nuclear export inhibitors (SINEs) indirectly target CRISPR-Cas9 mediated genome editing improving its specificity. The authors performed an EGFP reporter cell-based screening and obtained certain SINEs that showed the Cas9 inhibitory activity. The authors performed sufficient experiments to show the effect of SINEs on CRISPR-mediated base editing and proved that off-target effects were reduced. The indirect CRISPR-Cas9 inhibition with small-molecule inhibitors is of great interest to the field and might prove to be beneficial as a tool in biology or medicine. The manuscript is well written with clear methodological details. There are a few minor points that could be addressed to improve the manuscript.

Suppl. Figure 2a:

The fit of the data of KPT335 does not represent the data well. For example, the IC50 curve of KPT185 starts at a maximum EGFP activation of 20 %, while all other compounds show around 40 % maximum EGFP activation at low concentrations. Since the fit of the curve also does not represent the data well, the IC50 might be underestimated and the experiment should be repeated by the authors. Figure 1 should be updated with the newly tested IC50 values accordingly.

The inhibition% values of compound KPT330 on the genome-editing activity were not the same in suppl. Fig. 2b and 2c. Were the results from two different tests?

Fig. 2e:

XPO1 siRNA alone did not impact Cas9 mRNA transport. The authors assumed the reasons might be the residual XPO1 (Fig. 2c, d) or alternative transport pathway. However, if alternative transport pathway were part of the reason, then inhibition of Cas9 mRNA transport can be achieved by the alternative pathway. The authors should discuss this assumption with further clarification.

Fig. 2c:

There seems to be a significant difference on the knockdown levels of XPO1 between the KPT-330-treated group and the untreated group. What could be a potential explanation for this difference?

Fig.4b:

It is stated that "KPT330 at 0.5 μ M preferentially inhibits the on-target over out-of-window editing of A3A CBE at the pathogenic FBN1 site in a HEK293-based disease model of Marfan syndrome". While in the main text (line 174-176), the authors mentioned that "KPT185, KPT330, KPT335 and KPT8602 exhibited more inhibition toward out-of-window editing than on-target editing of BE3 or A3A CBEs". These are contradictory statements.

The authors show that the KPT330 could reduce off-target effects by CRISPR-Cas9. However, the potential off-target effects by KPT330 inhibiting nuclear export of other RNAs should be discussed.

The authors screened approximately 500 small-molecule compounds with irreversible warheads. Were all the irreversible warheads being Michael acceptors, or different other types were included?

Other points:

Line 41-42:

ABE and CBE should be defined in the main text in their first appearance: adenine base editor and

cytosine base editor

Line 44:

"...anti-CRISPR proteins (Acrcs) are most characterized CRISPR..." should be "...anti-CRISPR proteins (Acrcs) are the most characterized CRISPR..."

Line 151:

"...neither strong or weak ..." should be "...neither strong nor weak ...".

Line 128:

"along" should be changed to "alone".

Suppl Figure 3b:

the caption states "arrows indicate cleavage product" while there are no arrows in suppl. Fig. 3b; P-values should be reported.

Reviewer #2 (Remarks to the Author):

The manuscript by Cui and colleagues describes a strategy for down-regulating CRISPR-based genome modifying enzymes, which results in improved specificity in some cases. The manuscript supports its claims well. The mechanism of action is well supported: small molecule drugs (or SINEs) such as KPT330 can inhibit mRNA export from the nucleus, resulting in diminished amounts of editing effectors. However, because of the modest effects, it is unclear how much real-world impact this strategy will have. The manuscript is clear and the scientific conclusions are well-founded, but it is not apparent that this technology will be more useful than other options. If such advantages exist, they should be made more clear.

To make a case for the utility of this technology, it would be helpful to know how it compares to more established approaches, including "high fidelity" Cas9 constructs and/or inducible editor expression systems. The technology described in the present manuscript has the disadvantage of impacting all mRNAs in a given cell, which is not appealing. If the mechanism relies primarily on down-regulating the CRISPR gene expression, one might imagine similar results in an experiment wherein lower amounts of the CRISPR gene were delivered. In other words, if the goal is to down-regulate CRISPR, why not find a way to do so without down-regulating every other gene in the cell? Despite KPT330 being FDA approved for cancer patients, its substantial side effects should not be overlooked. Overall, these characteristics of the system may hinder its utility.

The materials & methods section is insufficient for a scientist to attempt to reproduce this work. This applies to both the main text and the supplement. The "EGFP reporter-based screen for CRISPR inhibitor" section does not describe what reagent or protocol was used for transfection. Plasmid pST-Cas9 is mentioned, but no description is provided; what does this plasmid encode? The supplement's section "Expression and purification of SpyCas9" is similarly flawed. "SpyCas9 WT" is described, but it is unlikely that the protein used was actually wild-type since WT Cas9 does not contain NLS sequences. All reagents and procedures must be described in sufficient detail for this work to be evaluated or interpreted.

A revised manuscript should do the following:

- (1) Expand discussion of (and comparison with) other strategies to accomplish the same goals of this technology.
- (2) Include suitable experimental details (either in the results section or the materials/methods) such that the work can be evaluated and potentially replicated.

Reviewer #3 (Remarks to the Author):

The study described in the manuscript entitled "Precision Genome and Base Editing by Regulating mRNA Nuclear Export Using Selective Inhibitors of Nuclear Export (SINEs)" by Professor Liu and colleagues tried to identify small molecules which inhibit CRISPR-Cas9 activity. The inhibitor of

CRISPR-Cas9 is important and useful to reduce non-specific mutation and recombination after it conducted a site-specific reaction. The authors conducted a screening of small molecules and found that the inhibitor of exportin 1 could suppress the CRISPR-Cas9-based genome editing. The conclusion is clearly described and the effect of KPT330 is convincing. However, there are several issues still not clarified in the manuscript. Overall, it was not clearly demonstrated whether KPT330 directly inhibits the function of XPO1 or reduce the amount of XPO1 by inhibiting its transcription. These two cases are completely different in understanding the inhibitory mechanism of SINE. The followings are the specific issues that I was concerned and should be clarified before the publication.

1. The authors cited ref49 and insisted that KPT330 inhibits XPO1 (CRM1). However, it seems that the ref49 does not describe the inhibition of CRM1 by KPT330. The authors should clarify this.
2. They demonstrated in Figure 2g that the treatment with KPT330 reduced the intracellular amount of XPO1. If this is the case, KPT330 does NOT inhibit the function of XPO1. It should be inhibiting transcription of XPO1 gene, or reducing the stability of XPO1 protein. It was not clear to me whether KPT330 inhibits the function of XPO1 or reduces the intracellular amount of XPO1. The authors should clarify this issue.
3. Related to the previous comment. The author conducted RNAi for XPO1 in Figure 2c and 2d to show the efficiency of siRNAs. The amount of XPO1 in control KD cells was not affected by 500 nM KPT330 (compare two red bars). However, this is obviously different from the result shown in Figure 2g, in which 500 nM KPT330 reduced the intracellular amount of XPO1. The author should clarify this.
4. In Figure 2a, they demonstrated that treatment of KPT330 and other SINE reduced the amount of cytoplasmic mRNA of Cas9 mRNA. I was wondering how they separated cytoplasmic and nucleoplasmic mRNAs, but could not find the detail method in Materials and Method section. The authors should describe the detailed procedure for the separation of mRNAs. In addition, it is necessary and highly recommended to quantify cytoplasmic mRNA fraction over the total (cytoplasm + nucleoplasm) fraction. Without measuring both nucleoplasmic and cytoplasmic mRNA levels, they cannot conclude that KPT330 inhibits nuclear transport (it may inhibit the transcription).
5. If KPT330 inhibits XPO1, they should see the same inhibitory effect against genome editing by other CRM1 inhibitor such as leptomycin B. They should show this.

Title: Precision Genome and Base Editing by Regulating mRNA Nuclear Export Using Selective Inhibitors of Nuclear Export (SINEs)COMMSBIO-21-0699

Authors: Cui YR, *et al.*

MS ID: COMMSBIO-21-0699

Reviewer #1 (Remarks to the Author)

Suppl. Figure 2a:

The fit of the data of KPT335 does not represent the data well. For example, the IC₅₀ curve of KPT185 starts at a maximum EGFP activation of 20 %, while all other compounds show around 40 % maximum EGFP activation at low concentrations. Since the fit of the curve also does not represent the data well, the IC₅₀ might be underestimated and the experiment should be repeated by the authors. Figure 1 should be updated with the newly tested IC₅₀ values accordingly.

RESPONSE-We thank the reviewer for pointing this out. It turned out that KPT185 was performed with a batch of EGFP reporter cells with too many passages. We thus repeated the experiments with a fresh batch of reporter cells. The new experiments yielded generally consistent results. We also repeated the curve fitting for KPT335 with the outlier data point removed. The IC₅₀s in SI Fig. 2 and Fig. 1 were updated accordingly.

The inhibition% values of compound KPT330 on the genome-editing activity were not the same in suppl. Fig. 2b and 2c. Were the results from two different tests?

RESPONSE-Yes, the results were obtained from two separate experiments and they were generally consistent.

Fig. 2e:

XPO1 siRNA alone did not impact Cas9 mRNA transport. The authors assumed the reasons might be the residual XPO1 (Fig. 2c, d) or alternative transport pathway. However, if alternative transport pathway were part of the reason, then inhibition of Cas9 mRNA transport can be achieved by the alternative pathway. The authors should discuss this assumption with further clarification.

RESPONSE-We thank the reviewer for raising this critical point. We have revised the Discussion section to further address this point.

Fig. 2c:

There seems to be a significant difference on the knockdown levels of XPO1 between the KPT-330-treated group and the untreated group. What could be a potential explanation for this difference?

RESPONSE-We thank the reviewer for raising this important question. Our results showed that KPT330 treatment upregulated XPO1 mRNA expression (Fig. 2d), though KPT330 induced degradation of XPO1 protein (Fig. 2g). In fact, similar differential effects of KPT330 on XPO1 protein and mRNA have been observed in previous studies (Breit *et al.*, 2014, *BMC Vet Res*; Tai *et al.*, 2014, *Leukemia*; Abdul Razak *et al.*, 2016, *J Clin Oncol*). These differential effects

are considered as the feedback mechanism of these SINE compounds.

Fig.4b:

It is stated that “KPT330 at 0.5 μ M preferentially inhibits the on-target over out-of-window editing of A3A CBE at the pathogenic FBN1 site in a HEK293-based disease model of Marfan syndrome”. While in the main text (line 174-176), the authors mentioned that “KPT185, KPT330, KPT335 and KPT8602 exhibited more inhibition toward out-of-window editing than on-target editing of BE3 or A3A CBEs”. These are contradictory statements.

RESPONSE-We apologize for the typos and the confusion they may cause. The text has been corrected to “KPT330 at 0.5 μ M preferentially inhibits the out-of-window over on-target editing of..”.

The authors show that the KPT330 could reduce off-target effects by CRISPR-Cas9. However, the potential off-target effects by KPT330 inhibiting nuclear export of other RNAs should be discussed.

RESPONSE-We thank the reviewer for raising this interesting question. We have revised the text in the Discussion sections to discuss the potential impact of SINEs on other types of RNAs.

The authors screened approximately 500 small-molecule compounds with irreversible warheads. Were all the irreversible warheads being Michael acceptors, or different other types were included?

RESPONSE-The initial library contains compounds with a variety of different functional groups. We have revised the text to further clarify this.

Other points:

Line 41-42:

ABE and CBE should be defined in the main text in their first appearance: adenine base editor and cytosine base editor.

RESPONSE-We thank the reviewer to point this out and have revised the main text.

Line 44:

“...anti-CRISPR proteins (Acrs) are most characterized CRISPR...” should be “...anti-CRISPR proteins (Acrs) are the most characterized CRISPR...”

RESPONSE-We thank the reviewer to point this out and have revised the main text.

Line 151:

“...neither strong or weak ...” should be “...neither strong nor weak ...”.

RESPONSE-We thank the reviewer to point this out and have revised the main text.

Line 128:

“along” should be changed to “alone”.

RESPONSE-We thank the reviewer to point this out and have revised the main text.

Suppl Figure 3b:

the caption states “arrows indicate cleavage product” while there are no arrows in suppl. Fig. 3b;

P-values should be reported.

RESPONSE-We thank the reviewer to point these out and have revised the text and figure accordingly.

Reviewer #2 (Remarks to the Author):

A revised manuscript should do the following:

(1) Expand discussion of (and comparison with) other strategies to accomplish the same goals of this technology.

RESPONSE-As suggested by the reviewer, we have expanded the Discussion section to compare SINEs with similar technologies. In addition, we have evaluated the effects of incorporation of high-fidelity Cas9 mutations on the specificity of base editors. Similar to some cases with Acrs, although high-fidelity Cas9 mutations largely limited the off-target activity, these effects were associated with great loss of on-target activity, thus rendering this approach with limited value for practical applications.

(2) Include suitable experimental details (either in the results section or the materials/methods) such that the work can be evaluated and potentially replicated.

RESPONSE-We have extensively revised the Methods section and Supplementary Information to include more experimental details.

Reviewer #3 (Remarks to the Author):

The followings are the specific issues that I was concerned and should be clarified before the publication.

1. The authors cited ref49 and insisted that KPT330 inhibits XPO1 (CRM1). However, it seems that the ref49 does not describe the inhibition of CRM1 by KPT330. The authors should clarify this.

RESPONSE- We have corrected the reference to clarify the KPT330 directly inhibits XPO1 by irreversible reaction.

2. They demonstrated in Figure 2g that the treatment with KPT330 reduced the intracellular amount of XPO1. If this is the case, KPT330 does NOT inhibit the function of XPO1. It should be inhibiting transcription of XPO1 gene, or reducing the stability of XPO1 protein. It was not clear to me whether KPT330 inhibits the

function of XPO1 or reduces the intracellular amount of XPO1. The authors should clarify this issue.

RESPONSE-We apologize for the confusion. KPT330 inhibits the function of XPO1 by reacting with its Cys528 residue (Mahipal *et al.*, 2016, *Pharmacol Ther*). This reaction can cause proteasome-dependent degradation of XPO1 protein (Tai *et al.*, 2014, *Leukemia*). The results in Fig. 2g were consistent with the previous studies. We have updated the reference and text to clarify this issue.

3. Related to the previous comment. The author conducted RNAi for XPO1 in Figure 2c and 2d to show the efficiency of siRNAs. The amount of XPO1 in control KD cells was not affected by 500 nM KPT330 (compare two red bars). However, this is obviously different from the result shown in Figure 2g, in which 500 nM KPT330 reduced the intracellular amount of XPO1. The author should clarify this.

RESPONSE-We thank the reviewer for raising this important question. Reviewer1 raised a similar concern on the opposite effects of KPT330 on the mRNA and protein levels of XPO1. In fact, similar differential effects of KPT330 on XPO1 protein and XPO1 mRNA have been observed in previous studies (Breit *et al.*, 2014, *BMC Vet Res*; Tai *et al.*, 2014, *Leukemia*; Abdul Razak *et al.*, 2016, *J Clin Oncol*). These differential effects are considered as the feedback mechanism of these SINE compounds. We have revised the text to clarify this.

4. In Figure 2a, they demonstrated that treatment of KPT330 and other SINE reduced the amount of cytoplasmic mRNA of Cas9 mRNA. I was wondering how they separated cytoplasmic and nucleoplasmic mRNAs, but could not find the detail method in Materials and Method section. The authors should describe the detailed procedure for the separation of mRNAs. In addition, it is necessary and highly recommended to quantify cytoplasmic mRNA fraction over the total (cytoplasm + nucleoplasm) fraction. Without measuring both nucleoplasmic and cytoplasmic mRNA levels, they cannot conclude that KPT330 inhibits nuclear transport (it may inhibit the transcription).

RESPONSE-We appreciate that the reviewer for raising these critical points. We have provided more experimental details in the Methods section. In addition, we quantify cytoplasmic mRNA fraction over total cellular mRNA and Fig. 2a was updated accordingly. The new results were generally consistent with the previous data and suggested that KPT330 inhibits nuclear transport rather than transcription.

5. If KPT330 inhibits XPO1, they should see the same inhibitory effect against genome editing by other CRM1 inhibitor such as leptomycin B. They should show this.

RESPONSE-We thank the reviewer for this helpful suggestion. We performed additional experiments to investigate the effects of leptomycin B on CRISPR-Cas9. As predicted by the reviewer, leptomycin B at 500 nM inhibited the on- and

off-target editing of CRISPR-Cas9 in human cells. Moreover, using an adjusted concentration of 100 nM, leptomycin B could improve the specificity of CRISPR-Cas9, as defined by the ratio between on-target and off-target activities. The new results are added as the new Supplementary Fig. 6.

Reviewers' comments:

Reviewer #1 (Remarks to the Author):

The authors have extensively revised the manuscript based on the peer review comments, provided detailed experiments procedure descriptions, and performed additional experiments. The current version of the manuscript is appropriate to be considered for publication by incorporating a few minor corrections listed below:

> Incorrect statement: "Small-molecule ..., ...but the application of these inhibitors to modulate CRISPR-Cas9-based genome- and base-editing tools is yet to be explored" because small-molecule Cas9 inhibitors have already been tested in a SpCas9-cytidine deaminase conjugate base-editing system: *Cell*, 2019, 177, 1067-1079, Figure S2S: 2-fold reduction in C to T conversion at 20 μ M for BRD7087 and BRD5779.

> Based on the data in revised Fig 1c, KPT335 showed \sim 15-fold more potent activity than KPT185 in the EGFP inhibition assay, while it seems that the inhibitory activity against on-target editing for the four genes between these two compounds are:

- either equivalent in most cases shown in Figure 1d and Figure S2c,
- or rather surprising that the most potent compound KPT335 showed the least inhibition against HBB editing.

Will be helpful if the author could propose an explanation for this.

> If KPT185, KPT276, KPT335 and KPT8602 are from the screened 500 compounds, the author should revise "an FDA-approved anticancer drug KPT330 displayed greatest potency for Cas9 inactivation", as compound KPT335 is the most potent (IC₅₀ 8 nM) based on the revised data.

> Fig 1b, the shared moiety was drawn in "E-configuration", while all the hits were drawn in "Z-configuration", should be clarified if all the screening compounds are "E-isomers" or if the "E-/Z-configuration" was evaluated for the identified hits.

> "We designed FOUR sgRNAs targeting to human EMX1 AAVS1 and HBB genes respectively" needs to be clarified (three or four genes?)

> Typo: "KPT330 appeared to have opposite effects on KPT....." revise as "...effects on XPO1..."

Reviewer #2 (Remarks to the Author):

The revised manuscript addresses the concerns I raised in my initial review. The revised manuscript also appears to address the concerns raised by the other reviewers. With this in mind, I have no further critical feedback.

Reviewer #3 (Remarks to the Author):

The points and concerns that I raised in the first draft have been solved except for the following one.

3. Related to the previous comment. The author conducted RNAi for XPO1 in Figure 2c and 2d to show the efficiency of siRNAs. The amount of XPO1 in control KD cells was not affected by 500 nM KPT330 (compare two red bars). However, this is obviously different from the result shown in Figure 2g, in which 500 nM KPT330 reduced the intracellular amount of XPO1. The author should clarify this.

RESPONSE-We thank the reviewer for raising this important question. Reviewer1 raised a similar concern on the opposite effects of KPT330 on the mRNA and protein levels of XPO1. In fact, similar differential effects of KPT330 on XPO1 protein and XPO1 mRNA have been observed in previous

studies (Breit et al., 2014, BMC Vet Res; Tai et al., 2014, Leukemia; Abdul Razak et al., 2016, J Clin Oncol). These differential effects are considered as the feedback mechanism of these SINE compounds. We have revised the text to clarify this.

>The authors explained the discrepancy between mRNA level and protein level. However, to my understanding, both figure 2d and 2g are western blot, which indicates protein level and not mRNA level. I still see the discrepancy between these two figures: 500 nM KPT330 reduced the 'protein level' of XPO1 in Figure 2g but not in Figure 2d. The authors still need to clarify this issue.

Reviewers' comments:

Reviewer #1 (Remarks to the Author):

The authors have extensively revised the manuscript based on the peer review comments, provided detailed experiments procedure descriptions, and performed additional experiments. The current version of the manuscript is appropriate to be considered for publication by incorporating a few minor corrections listed below:

> Incorrect statement: “Small-molecule ..., ...but the application of these inhibitors to modulate CRISPR-Cas9-based genome- and base-editing tools is yet to be explored” because small-molecule Cas9 inhibitors have already been tested in a SpCas9-cytidine deaminase conjugate base-editing system: *Cell*, 2019, 177, 1067-1079, Figure S2S: 2-fold reduction in C to T conversion at 20 μ M for BRD7087 and BRD5779.

Response: We thank the reviewer to point this out and have revised the text to better describe the results in previous studies.

> Based on the data in revised Fig 1c, KPT335 showed ~15-fold more potent activity than KPT185 in the EGFP inhibition assay, while it seems that the inhibitory activity against on-target editing for the four genes between these two compounds are:

- either equivalent in most cases shown in Figure 1d and Figure S2c,
- or rather surprising that the most potent compound KPT335 showed the least inhibition against HBB editing.

Will be helpful if the author could propose an explanation for this.

Response: We thank the reviewer to raise this question. Indeed, based on the EGFP reporter assay the IC₅₀ of KPT335 was 15-fold lower than that of KPT185, while in Fig.1d and Fig.S2c, KPT335 and KPT185 had similar inhibitory activity. A simple explanation is that the conditions of these experiments were different. Fig.1c and Fig.1d or Fig.S2c were carried out at different genomic sites (EGFP in Fig.1c, HBB in Fig.1d and AAVS, HBB, EMX1-2 in Fig.S2c) with different concentrations of compound (titrating doses in Fig.1c, 0.5 μ M in Fig.1d and 10

μM in Fig.S2c). We believe that the inhibitory activity of SINEs toward CRISPR-Cas9 is dependent on both the genomic sites and the concentrations of compounds, as observed with other experiments in the present study. We have revised the text to better clarify the conditions of each experiment.

> If KPT185, KPT276, KPT335 and KPT8602 are from the screened 500 compounds, the author should revise “an FDA-approved anticancer drug KPT330 displayed greatest potency for Cas9 inactivation”, as compound KPT335 is the most potent (IC_{50} 8 nM) based on the revised data.

Response: We have revised the text to keep the whole manuscript consistent.

> Fig 1b, the shared moiety was drawn in “E-configuration”, while all the hits were drawn in “Z-configuration”, should be clarified if all the screening compounds are “E-isomers” or if the “E-/Z-configuration” was evaluated for the identified hits.

Response: We apologize for the confusion. The identified hits contain both E- and Z-configurations (Table S1). We have revised Fig.1b to indicate this.

> “We designed FOUR sgRNAs targeting to human EMX1 AAVS1 and HBB genes respectively” needs to be clarified (three or four genes?)

Response: There were four sgRNAs targeting to three genes. We have revised the text to clarify this.

> Typo: “KPT330 appeared to have opposite effects on KPT.....” revise as “...effects on XPO1...”

Response: We thank the reviewer to point this out and have corrected this typo.

Reviewer #2 (Remarks to the Author):

The revised manuscript addresses the concerns I raised in my initial review. The revised manuscript also appears to address the concerns raised by the other reviewers. With this in mind, I have no further critical feedback.

Reviewer #3 (Remarks to the Author):

The points and concerns that I raised in the first draft have been solved except for the following one.

3. Related to the previous comment. The author conducted RNAi for XPO1 in Figure 2c and 2d to show the efficiency of siRNAs. The amount of XPO1 in control KD cells was not affected by 500 nM KPT330 (compare two red bars). However, this is obviously different from the result shown in Figure 2g, in which 500 nM KPT330 reduced the intracellular amount of XPO1. The author should clarify this.

RESPONSE-We thank the reviewer for raising this important question. Reviewer1 raised a similar concern on the opposite effects of KPT330 on the mRNA and protein levels of XPO1. In fact, similar differential effects of KPT330 on XPO1 protein and XPO1 mRNA have been observed in previous studies (Breit et al., 2014, BMC Vet Res; Tai et al., 2014, Leukemia; Abdul Razak et al., 2016, J Clin Oncol). These differential effects are considered as the feedback mechanism of these SINE compounds. We have revised the text to clarify this.

>The authors explained the discrepancy between mRNA level and protein level. However, to my understanding, both figure 2d and 2g are western blot, which indicates protein level and not mRNA level. I still see the discrepancy between these two figures: 500 nM KPT330 reduced the 'protein level' of XPO1 in Figure 2g but not in Figure 2d. The authors still need to clarify this issue.

Response: We apologize for the confusion. Fig.2d was actually the mRNA expression, determined by RT-qPCR. We have revised the figure legend to clearly indicate this.

REVIEWERS' COMMENTS:

Reviewer #3 (Remarks to the Author):

The issues of my concern has been clarified in the revised manuscript. I think the manuscript is now ready for publication.